# Heparin-Induced Changes of Vascular Endothelial Growth Factor (VEGF_165_) Structure

**DOI:** 10.3390/biom13010098

**Published:** 2023-01-03

**Authors:** Ekaterina L. Nemashkalova, Marina P. Shevelyova, Andrey V. Machulin, Dmitry D. Lykoshin, Roman S. Esipov, Evgenia I. Deryusheva

**Affiliations:** 1Institute for Biological Instrumentation, Pushchino Scientific Center for Biological Research of the Russian Academy of Sciences, Pushchino 142290, Russia; 2Skryabin Institute of Biochemistry and Physiology of Microorganisms, Pushchino Scientific Center for Biological Research of the Russian Academy of Sciences, Pushchino 142290, Russia; 3Shemyakin-Ovchinnikov Institute of Bioorganic Chemistry, Russian Academy of Sciences, Moscow 117997, Russia

**Keywords:** vascular endothelial growth factor, heparin, protein aggregation, differential scanning calorimetry

## Abstract

Vascular endothelial growth factor-A (VEGF-A), a secreted homodimeric glycoprotein, is a critical regulator of angiogenesis in normal and pathological states. The binding of heparin (HE) to VEGF_165_ (the major form of VEGF-A) modulates the angiogenesis-related cascade, but the mechanism of the observed changes at the structural level is still insufficiently explored. In the present study, we examined the effect of HE on the structural and physicochemical properties of recombinant human VEGF_165_ (rhVEGF_165_). The HE binding results in an increase of hydrophobic surface exposure in rhVEGF_165_ without changes in its secondary structure. Differential scanning calorimetry measurements for intact and HE-bound rhVEGF_165_ reveals the absence of any pronounced thermally induced transitions in the protein in the temperature range from 20 to 100 °C. The apolar area increase during the heparin binding explains the pronounced HE-induced oligomerization/aggregation of rhVEGF_165_, as studied by chemical glutaraldehyde cross-linking and dynamic light scattering. Molecular modeling and docking techniques were used to model the full structure of dimeric VEGF_165_ and to reveal putative molecular mechanisms underlying the function of the VEGF_165_/HE system. In general, the results obtained can be a basis for explaining the modulating effect of HE on the biological activity of VEGF-A.

## 1. Introduction

Vascular endothelial growth factors (VEGFs) are a family of secreted signaling proteins playing a critical role in vasculo- and angiogenesis. Both processes are essential for physiologic homeostasis and the pathogenesis of tumor growth, metastasis and ophthalmic diseases [1,2,3]. The VEGF levels in the plasma and serum of healthy people are 19–47 pg·mL^−1^ and 119–238 pg·mL^−1^, respectively, while a marked increased concentration have been observed in various types of cancer (rev in [4]). The VEGF gene family includes VEGF-A, VEGF-B, VEGF-C, VEGF-D, virally encoded VEGF-E, and placenta growth factor, differing in their expression pattern, biological functions and receptor specificity [5].

VEGF proteins, belonging to the large PDGF (platelet-derived growth factor) family (SCOP ID 4003427), are members of the cystine knot growth factor superfamily (SCOP ID: 3000846). VEGF-A plays a dominant role in the regulation of angiogenesis. It binds to two tyrosine kinase receptors: vascular endothelial growth factor receptor 1 (VEGFR1) and 2 (VEGFR2) [6,7,8]. VEGF binding to VEGFRs promotes a dimerization of the intracellular domain of VEGFRs, which subsequently leads to the activation of a cascade of downstream signaling molecules [9]. For efficient signal transduction, VEGF requires the binding of co-receptors, neuropilin-1 [10,11] and heparin (HE) or heparan sulfate (HS) [12]. HE and HS are glycosaminoglycans (GAGs), which are anionic linear periodic polysaccharides composed of repetitive disaccharide units containing an uronic acid and an amino sugar ring. GAGs are located on cell surfaces and in the extracellular matrix and participate in diverse signaling processes via interactions with their protein targets [13,14]. The range of plasma HE concentration in clinically normal individuals is 1–2.4 mg∙L^−1^ [15].

The alternative splicing of VEGF-A gives rise to peptides of 121 (VEGF_121_), 165 (VEGF_165_), 189 (VEGF_189_), and 206 (VEGF_206_) amino acids. These variants share a common N-terminal receptor-binding domain (RBD) of 115 residues and have a similar affinity for VEGFR2 [16]. X-ray crystallography of the RBD (residues 8-109) has shown that VEGF forms an antiparallel homodimer covalently linked by two disulfide bridges between Cys-51 and Cys-60 with a central four-stranded β-sheet and cystine knot motif within the monomer [17,18]. The key difference between spliced variants of VEGF is their ability to bind HE and HS proteoglycans [19]. The affinity of the VEGF forms to HE is increased in the following order: VEGF_165_, VEGF_189_, and VEGF_206_, whereas VEGF_121_ does not bind heparin [19].

VEGF-A forms capable of binding HE contain the same C-terminal heparin-binding domain (HBD). Nuclear magnetic resonance (NMR) spectroscopy of VEGF_165_ HBD (PDB:1VGH, NMR, 55 residues) revealed two subdomains, each containing two disulfide bridges and a short two-stranded antiparallel β-sheet; the carboxy-terminal subdomain also contains a short α helix [20]. The loss of the carboxyl-terminal domain, whether due to proteolysis or alternative splicing, correlates with a substantial decrease in endothelial cell mitogenic activity of VEGF [21]. The isolated HBD was shown to have high affinity towards HE, comparable to that observed for the full VEGF_165_ [20,21,22]. Surface electrostatic potential calculations revealed that positively charged residues on one side of the HBD surface could make up a potential binding site for the negatively charged HE. The GAGs binding site in the HBD is mapped out by Lys and Arg residues located towards the C-terminus of the domain [11,20], but an experimental structure for HBD in complex with HE is not available. Additionally, three-dimensional structure that includes both RBD and HBD has not yet been obtained either due to the challenges of the crystallization procedure or to the highly flexible linker that connects them. Several models were proposed for the full VEGF: some of them are able to bind VEGFR2, for others, such binding is sterically hindered. In some models, HE binds to VEGF through its HBD and is able to redistribute the conformational ensemble of the protein, whereas the binding of HE to RBD is not favorable. It is also assumed that HE binding could induce the formation of a sandwich structure, where one HE molecule is symmetrically bound by two HBDs [11,23,24].

The apparent dissociation constant (*K_D_*) value for the HE–VEGF-A complex is 11–80 nM [25,26]. HE binding is relevant to VEGF activity within endothelial cells. The cell surface-associated heparin-like molecules were shown to be required for the interaction of VEGF with its cell surface receptors [12]. The amount of VEGF_165_ bound to VEGFR2 in the presence of HE increases more than three-fold, while the apparent affinity of protein for the receptor does not change [21]. At the same time, the affinity of VEGF_165_ to VEGFR1 decreases in the presence of heparin, but the remaining binding results in more efficient kinase activation [27]. The difference between HE-modulating actions on VEGFRs may be explained by the presence of heparin-binding peptides in the extracellular domain of VEGFR2 but not VEGFR1 [28]. Notably, VEGF_110_ (the form of VEGF_165_, which lacks HBD) maintains the ability of the native protein to bind to the VEGF receptors, but the endothelial cell mitogenic potency of homodimeric VEGF_110_ is decreased more than 100-fold compared to that of VEGF_165_ [21].

VEGF binding to Avastin (commercial name of bevacizumab), the first humanized anti-VEGF monoclonal antibody approved by the US Food and Drug Administration for the first-line treatment of metastatic colorectal cancer [29], is significantly enhanced in the presence of HE, with a maximal effect (>2-fold) being observed in the presence of 0.1 μg·mL^−1^ HE [30]. Moreover, the reduced Avastin–VEGF binding at an acidic pH is rescued by HE [30]. Ko et al. have showed that VEGF_189_ exhibits a high affinity to HE, and is preferentially localized in cancer cell-derived small extracellular vesicles (sEVs) via heparin-binding. sEV-associated VEGF is highly stable and is not neutralized by the therapeutic VEGF antibody bevacizumab, raising the possibility that elevated levels of sEV-VEGF contribute in part to the resistance of tumors to bevacizumab [31].

Despite the plethora of works devoted to the effect of HE/HS on the biological activity of VEGF, the mechanism of the observed changes at the structural level is still insufficiently explored. In the present study, we examined the effect of HE on the structural and physico-chemical properties of human VEGF_165_, the most abundant and biologically active form of VEGF. In this work, a modeling approach in combination with circular dichroism (CD) spectroscopy, dynamic light scattering (DLS) and differential scanning calorimetry (DSC) was applied to characterize the putative full-length VEGF_165_ structure and to reveal the impact of HE binding on it.

## 2. Materials and Methods

### 2.1. Materials

Recombinant human VEGF_165_ produced in *E. coli* was obtained from the Institute of Bioorganic Chemistry of the Russian Academy of Sciences. Recombinant human VEGF_165_ produced in Chinese Hamster Ovary (CHO) cells (#PSG010-10) was purchased from SCI-store (Moscow, Russia). Unfractioned sodium heparin from the intestinal mucosae of pig (HE) 5000 IU/mL was from Velpharm (Kurgan, Russia). Sodium chloride, sodium di-hydrogen phosphate 2-hydrate and sodium dodecyl sulfate (SDS) were from Panreac Applichem (Darmstadt, Germany). Ultra-grade Tris was purchased from VWR Life Science AMRESCO (Vienna, Austria). Bis-ANS (4,4′-dianilino-1,1′-binaphthyl-5,5′-disulfonic acid), grade II glutaraldehyde, and Coomassie R-250 were from Merck (Darmstadt, Germany). All buffers and other solutions were prepared using ultrapure water.

The freeze-dried recombinant human VEGF_165_ (rhVEGF_165_) and HE were dialyzed exhaustively against a suitable buffer (cutoff 12–14 kDa and 1 kDa were used for rhVEGF-165 and HE, respectively) before measurements. Protein concentrations were measured spectrophotometrically using molar extinction coefficients at 280 nm of 13,920 M^−1^cm^−1^ calculated according to previous work [32]. The concentration of water stock solution of bis-ANS was evaluated using the molar extinction coefficient at 385 nm of 16,790 M^−1^cm^−1^ [33]. The mass concentration of the HE stock solution after dialysis was calculated from the freeze-dried mass of HE. The average molar concentration was estimated using the average molecular mass of HE 13.8 kDa; the molar concentration of disaccharide residues was calculated using their molecular mass. The average molecular mass of HE was estimated using the average hydrodynamic diameter measured by DLS method, using a set of linear polymers of different molar masses (from 8 to 400 kDa) as a reference. The evaluated diameter value agreed with the literature data for similar pharmaceutical HEs of the same origin [34].

### 2.2. Circular Dichroism Studies

Far-UV CD measurements were performed using a J-810 spectropolarimeter (JASCO, Inc., Tokyo, Japan) equipped with a Peltier-controlled cell holder. A quartz cell with pathlength of 1 mm was used. Protein concentration was 1.8–1.9 µM. HE was added up to an equimolar ratio to the protein. The contribution of buffer (pH 7.4, 10 mM Na-phosphate, 50 mM NaCl) in the absence/presence of HE was subtracted from the experimental spectra. Estimations of the content of the secondary structure elements were made using the CDPro software package with SDP48 and SMP56 reference protein sets [35].

### 2.3. Scanning Calorimetry Measurements

The scanning microcalorimetry studies were carried out on a Nano DSC microcalorimeter (TA Intruments Inc., New Castle, DE, USA) at a 1 K·min^−1^ heating rate and excess pressure of 4 bars (pH 7.4, 10 mM Na-phosphate, 50 mM NaCl) in the presence/absence of 3.8 µM of HE (88.2 µM of disaccharide units). Protein concentrations were 1.0–1.4 µM. The protein specific heat capacity (*c_p_*) was calculated as described by Privalov and Potekhin [36]. The partial molar volume and specific heat capacity of a fully unfolded protein were estimated according to Häckel et al. [37,38] and Makhatadze and Privalov [39,40].

### 2.4. Fluorescence Studies

Fluorescence spectra were measured using a Cary Eclipse spectrofluorometer (Varian Inc., Palo Alto, CA, USA), equipped with a Peltier-controlled cell holder, using quartz cells at 25 °C. Buffer conditions were the following: pH 7.4, 10 mM Na–phosphate, 50 mM NaCl. The rhVEGF_165_ and HE concentration was 25 µM (or 580 µM of disaccharide units). Fluorescence of bis-ANS (2.5 µM) was excited at 385 nm. The emission band width was 5 nm. All spectra were fitted to log-normal curves [41] using LogNormal software (IBI RAS, Pushchino, Russia). Fluorescence emission maximum positions and quantum yields were obtained from these fits.

### 2.5. Dynamic Light Scattering Studies

DLS measurements were carried out using a Zetasizer Nano ZS (Malvern Instruments Ltd., Malvern, UK) system. The backscattered light from a 4 mW He-Ne laser 632.8 m was collected at an angle of 173°. The VEGF concentration was 11 µM (256 µM of disaccharide units). Buffer conditions were the following: pH 7.4, 10 mM Na–phosphate, 50 mM NaCl. Sample temperature was 25 °C. The acquisition time for the single autocorrelation function was 150 s. The resulting autocorrelation functions are averaged values from three measurements. The volume-weighted size distributions were calculated using the following parameters for the sodium phosphate buffer: refractive index of 1.333 measured with RL3 refractometer (PZO, Warszawa, Poland); the viscosity value for sodium phosphate buffer *η* = 0.882 mPa∙s measured using micro-rheology method with a water suspension of standard latex nanoparticles. The hydrodynamic diameter for natively unfolded protein was calculated from the correlation equation for hydrodynamic volume *V_h_* log(*V_h_*) = *f*(log(*N*)), where *N* = 330 is the number of amino acid residues of dimeric VEGF_165_ [42].

### 2.6. Chemical Crosslinking of Proteins

Crosslinking of rhVEGF_165_ (25 μM) with 0.02% glutaraldehyde was performed at a pH of 7.4, with 10 mM Na–phosphate and 50 mM NaCl, in the absence/presence of 25 µM HE (or 580 µM of disaccharide units) as mainly described in [43]. The samples were subjected to sodium dodecyl sulfate–polyacrylamide gel electrophoresis (SDS-PAGE) under non-reducing conditions (5% concentrating and 12% resolving gels).

### 2.7. Modeling the Full Structure of Human VEGF_165_ and Its Complex with HE

The tertiary structures of the RBD and HBD domains of VEGF_165_ were taken as the basis for modeling (PDB entries 2VPF and 1VGH, respectively). The homodimeric form of the VEGF_165_ protein was predicted by the AlphaFold server (https://alphafold.ebi.ac.uk/, accessed on 1 November 2022) in the multimer model option based on the VEGF_165_ sequence (UniProt ID: P15692-4). The signal peptide was excluded from the protein amino acid sequence for the prediction. I-TASSER server (https://zhanggroup.org/I-TASSER/, accessed on 1 November 2022) predicted the monomeric subunits of VEGF_165_ (HBD, linker, HBD). The alignment, refinement and visualization of 3D structures were performed using the Pairwise Structure Alignment RCSB PDB online service (https://www.rcsb.org/alignment, accessed on 1 November 2022), Stride server (http://webclu.bio.wzw.tum.de/cgi-bin/stride/stridecgi.py, accessed on 1 November 2022) and PyMOL v. 2.5 (https://pymol.org, accessed on 1 November 2022). The calculation of the content of secondary structure elements in the VEGF_165_ model, calculation of the surface areas (solvent-accessible surface area, polar and apolar surface area) and the measurement of linear parameters were carried out using scripts in PyMOL v. 1.6 (https://pymol.org, accessed on 1 November 2022).

A total of 25 models of the VEGF_165_–HE complex were generated using ClusPro docking server [44] in the heparin-binding mode [45]. The contact residues in the docking models of VEGF_165_–HE complex were calculated using Python 3.5 programming language (implemented in PyCharm Community Edition 2020 development environment), Matplotlib Python plotting library and NumPy numerical mathematics extension. The numbering of the contact residues is according to the protein data bank (PDB) entries. The tertiary structure models were drawn with the molecular visualization system PyMOL v. 1.6.

## 3. Results and Discussion

### 3.1. The Effect of Heparin on the Secondary and Tertiary Structure of VEGF_165_

It has been previously shown that the secondary structure of VEGF_165_ consists predominantly of unordered (43%) and β-sheet (32%) parts [26]. Since the functional properties of VEGF_165_ expressed in pro- and eukaryotes are identical, we have compared the secondary structure of the proteins derived from various sources. An analysis of far-UV CD spectra for intact rhVEGF_165_, expressed in eukaryotic (CHO) and prokaryotic (*E. coli*) cells, reveals a similar content of these elements in the secondary structure (Table 1). An addition of equimolar amount of HE does not change the secondary structure of rhVEGF_165_ (Table 1). Notably, the two-fold increase of HE concentrations also does not change the distribution of the secondary structure fractions. Early, Wijelath et al. showed for VEGF_165_ that HE induces a partial transition of β-sheets to α-helices, but the source and preparation method of the protein, as well as buffer conditions of the measurements were not specified [26], which makes it difficult to compare the results.

To assess the heparin-induced alterations in rhVEGF_165_ tertiary structure, we used DSC and hydrophobic probe bis-ANS fluorescence methods. The thermogram for rhVEGF_165_ in the absence of heparin has no cooperative peaks characteristic of the protein thermal denaturation (Figure 1A) in the temperature range from 20 to 100 °C. The decrease in the thermogram at temperatures above 100 °C seems to be caused by a protein aggregation. The high level of specific heat capacity for VEGF-A_165_ and the absence of cooperative transitions in the wide temperature region suggest the absence of a rigid tertiary structure in this protein. The addition of a two-fold amount of HE has practically no effect on the thermal behavior of the protein, but the increase of absolute heat capacity up to the heat capacity of fully unfolded VEGF evidences an increase of relative proportion of solvent accessible apolar areas.

Since the methods used hereinafter in our work (spectrofluorimetry, DLS, chemical crosslinking) require a large amount of protein, in some experiments we used rhVEGF_165_ expressed in *E. coli*. The bacterial expression systems are well-suited to produce significant amounts of rhVEGF_165_ with secondary structures identical to that of the protein expressed in eukaryotic cells (Table 1). Moreover, N-linked glycosylation of rhVEGF_165_ was shown not to affect its functional properties [21,46,47].

The results of spectrofluorimetric monitoring of hydrophobic probe bis-ANS interactions with rhVEGF_165_ in the absence/presence of HE are shown in Figure 1B. Interactions with nonpolar solvent-exposed cavities of proteins result in an increase of fluorescence quantum yield and a blue shift of the bis-ANS fluorescence spectrum maximum [48]. Heparin binding to rhVEGF_165_ is accompanied by a 28% increase of bis-ANS quantum yield with a slight (2 nm) red shift of the spectrum, apparently reflecting heparin-induced exposure of the hydrophobic surface of rhVEGF_165_. Similar effect was found early for ANS binding to bovine serum albumin in the presence of heparin and was explained by the formation of protein aggregates [49]. It can be assumed that the process of HE binding by two HBDs domains may promote a VEGF rearrangement with the opening of the cavity leading to an increase in the area of hydrophobic surfaces. The slight red shift in probe fluorescence may suggest that HE-binding to rhVEGF_165_ induces the filling of additional sites for bis-ANS, in which the polarity of the probe environment is increased.

### 3.2. HE-Dependent Changes in the Quaternary Structure of rhVEGF_165_

Since the observed HE-induced structural alteration of rhVEGF_165_ may partly be a consequence of changes in oligomeric state of the proteins, we have examined this possibility using chemical crosslinking of the protein with 0.02% glutaraldehyde (37 °C for 1 h). We did not use reducing agents in the following SDS-PAGE analysis since all SH groups in the VEGF_165_ homodimer are included into the intermolecular disulfide bridges. About a third of intact rhVEGF_165_ exists as dimers and third of the protein could not penetrate the 12% resolving gels (high-molecular weight forms) (Figure 2A). Heparin binding to rhVEGF_165_ leads to approximately a two-fold increase of high-molecular-weight forms and an almost complete disappearance of dimers.

DLS spectroscopy confirmed HE-induced oligomerization of rhVEGF_165_ at 25 °C, which is clearly seen in the increase in its hydrodynamic diameter, *D_h_* (Figure 2B). Reference rhVEGF_165_ exhibits a size distribution of *D_h_* = (6.5 ± 1.3) nm, which is satisfactory with that calculated for such a protein in the natively unfolded state [42], and the addition of equimolar quantity of HE causes a shift of the main scattered peak towards higher *D_h_* values: (12.8 ± 1.4) nm together with aggregates formation detected at nearly 1 μm.

Heparin-induced aggregation of various proteins has been shown earlier (reviewed in [49]). Study of this process using bovine serum albumin as a model protein has revealed the downhill polymerization mechanism of the HE-induced protein aggregation [49]. Moreover, HS presents in most, if not all, intra- and extracellular protein aggregates that accumulate in neurodegenerative diseases, including those made of Aβ, tau, and α-synuclein (reviewed in [50]).

### 3.3. Model of VEGF_165_–HE Complex

Since the three-dimensional structure of full VEGF_165_ is still not clear, we used AlphaFold server for modeling the structure of homodimeric VEGF_165_, Figure 3A. Two structural domains (RBD and HBD) are clearly seen in the figure. Moreover, the spatial arrangement of monomeric subunits during the dimerization contributes to the formation of a structural cavity in the center of the molecule. The linear parameters of the VEGF_165_ homodimer model 65 × 73 × 81 Å agree with the *D_h_* value. The secondary structure content calculated from the model using PyMOL (8.5%, 36.5% and 55 % for α-helices, β-sheets, and unordered structure, respectively) is consistent with the experimental data (Table 1). Taken together, these data indicate that our model adequately reflects the structure of VEGF_165_.

In accordance with the literature data, the HBD of homodimeric VEGF would be preferred for HE binding, whereas a negative electrostatic potential of RBD is not prone to HE interactions [20]. This finding is partially confirmed by the docking results obtained for VEGF–HE models generated using the ClusPro docking server [44]: 25 top ranked solutions indicate that HE binds predominantly to HBD (Figure 3B). It should be noted that the model proposed in our work differs from the one proposed in [23] by a more compact arrangement of the HBDs. However, for some models, the binding sites include residues from RBD. Notably, our proposed model allows binding both by one HBD and by two HBDs (Figure 3B). Out of 25 models, only in 6 models HE interacts with only one chain of the homodimer VEGF_165_. The following VEGF_165_ residues contribute most to the HE binding: Lys48, Asn62, Glu64, Glu67, Gly65, and His86 from RBD, and Arg156, Arg159, Lys162, and Arg164 from HBD. Some of these residues (Lys48, His86) are part of the receptor-binding sites (PDB IDs 1FLT and 3V2A for VEGFR1 and VEGFR2, respectively). Notably, the bevacizumab-binding site in VEGF_165_(PDB ID 6BFT) overlaps with some model VEGF_165_-HE sites (His86, Gln87, Gly88, Gln89). In several models, contacts of HE with both linkers (between RBD and HBD) belonging to two chains were found.

The PyMOL program allows us to evaluate the solvent-accessible and buried areas in VEGF_165_ and its complex with HE. SAS (solvent-accessible surface) polar and apolar areas are equivalent to the contact area between polar or non-polar amino acid residues and a solvent, while the buried surface area corresponds to the amino acid area excluded from the solvent interaction [51]. Since the exterior amino acids participate in protein–protein and protein–solvent interactions, a change in SASap (solvent-accessible apolar area) and SASpol (solvent accessible polar area) together with BSA (area buried from the solvent) correlates with driving forces in protein folding and conformational rearrangement upon binding [52]. It was shown that for intrinsically disordered proteins the higher the flexibility of the protein when compared to a folded protein of the same molecular mass, the deeper the conformational changes upon binding [52,53,54]. SAS of homodimeric VEGF_165_ is composed predominantly of polar amino acids, forming SASpol. Notably, the SASpol is almost three times more than SASap while normal area ratio is about 55% of SASap and 45% of SASpol [55]. Upon HE binding, a part of polar positively charged amino acid residues becomes shielded, so SASpol’s relative contribution to SAS decreases (Figure 3B and Table 2), which is qualitatively agreed with calorimetric and bis-ANS fluorescence data (Figure 1).

## 4. Conclusions

The effect of HE and its derivatives on the biological activity of VEGF may be partly explained by the HE-induced structural changes of VEGF revealed in our work. The experimental and modeling results obtained in our study allow us to consider VEGF_165_ as an intrinsically disordered protein with abnormally large solvent-accessible surface areas. The interaction with HE does not affect the secondary structure of VEGF_165_, but increases its hydrophobic areas exposed to the solvent and changes its quaternary structure. The aggregation of rhVEGF_165_ can partly explain the increase of VEGFR2-bound VEGF_165_ in the presence of HE [21]. Moreover, the predicted HE-binding site of VEGF shares several amino acids with bevacizumab and receptor-binding sites, which may explain the HE modulation of VEGF_165_’s interaction with antibodies/VEGFRs [30]. In particular, the HE-shielding of positively charged residues of VEGF_165_ in the VEGFR1 binding site could result in a decrease in protein–receptor binding efficiency, found earlier [27]. The presented modeling data allow us to propose a molecular mechanism for the participation of HE in the VEGF binding processes. In the absence of HE, VEGF_165_ apparently can take on several conformations, which differ in the mutual orientation of HBDs due to flexible linkers. This flexibility allows the binding of VEGF_165_ to the receptor and thus the activation of the signaling pathway [6]. Further investigations of these HE-mediated mechanisms could be of practical use for the production of novel biomaterials with well-understood and controlled properties for their implementation in tissue regeneration therapies.

## Figures and Tables

**Figure 1 biomolecules-13-00098-f001:**
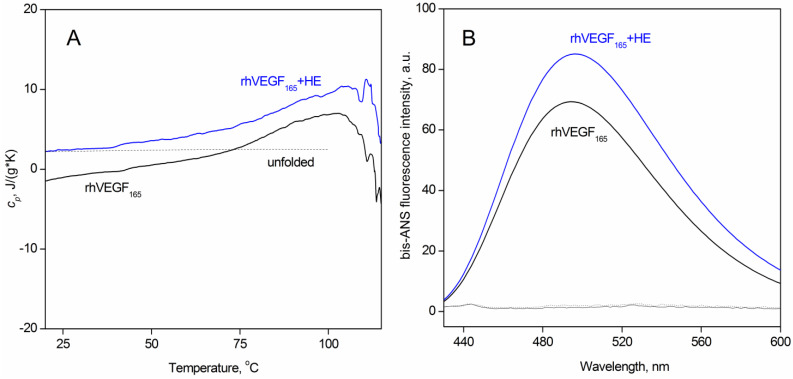
(**A**) Temperature dependencies of specific heat capacity for rhVEGF_165_ (1 µM) in the absence (blue line) and the presence of 3.8 µM HE (black line). Grey dashed line corresponds to unfolded VEGF_165_, calculated according to [38]; (**B**) Fluorescence spectra of bis-ANS (2.5 μM) in the absence (black line) and in the presence of heparin (blue line) in the solution containing 2.5 μM rhVEGF_165,_ monitored by fluorescence emission spectrum of the dye at 25 °C. Thin solid and dashed lines correspond to spectra of bis-ANS in buffer in the presence (grey line) or absence (dashed line) of HE. Buffer conditions: 10 mM Na-phosphate, 50 mM NaCl, pH 7.4.

**Figure 2 biomolecules-13-00098-f002:**
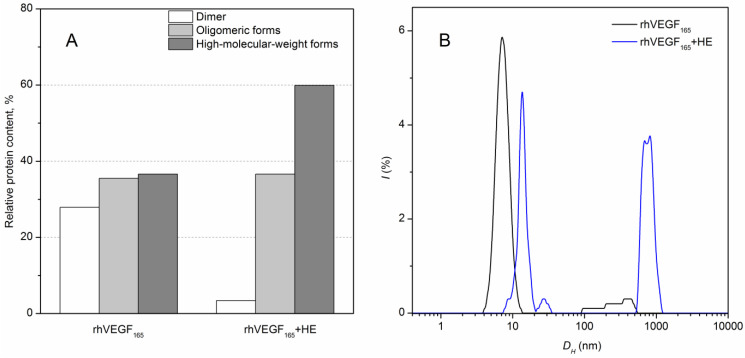
HE-dependent changes in quaternary structure of rhVEGF_165_ (10 mM Na-phosphate, 50 mM NaCl, pH 7.4). (**A**) Effects of HE (25 µM) on distribution of rhVEGF_165_ (25 μM) multimeric forms at 37 °C, as judged from glutaraldehyde crosslinking experiments at 37 °C; (**B**) DLS data on size distribution (hydrodynamic diameter *D_h_*) of rhVEGF_165_ (10 µM) in solutions in the absence (black line) and presence of 11 µM HE (blue line) at 25 °C.

**Figure 3 biomolecules-13-00098-f003:**
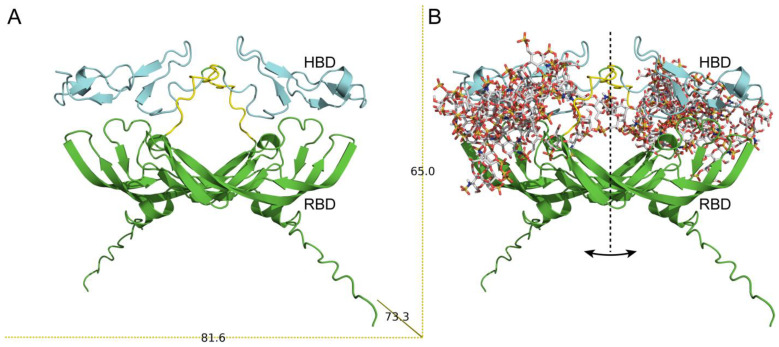
(**A**) The model of homodimeric VEGF_165_. RBD and HBD domains are highlighted by green and cyan, respectively; (**B**) Overlay of the 25 representative models of the VEGF_165_–HE complexes calculated using ClusPro docking server [44]. HE is shown in the stick representation.

**Table 1 biomolecules-13-00098-t001:** The secondary structure fractions estimated from the CD data using CDPro software package [35] for rhVEGF_165_ expressed in CHO or *E. coli* at 25 °C (pH 7.4, 10 mM Na–phosphate, 50 mM NaCl). Protein concentration was 1.8–1.9 µM. HE was added up to an equimolar ratio to the protein.

Protein	α-Helices, %	β-Sheets, %	Turns, %	Unordered Structure, %
rhVEGF_165_-CHO	8.8 ± 3.5	30.2 ± 4.5	22.7 ± 2.8	38.4 ± 4.8
rhVEGF_165_-CHO + HE	8.1 ± 2.4	31.9 ± 2.9	22.3 ± 2.1	37.7 ± 3.2
rhVEGF_165_-*E. coli*	8.7 ± 2.7	29.4 ± 2.8	22.3 ± 2.7	39.6 ± 5.3
rhVEGF_165_-*E. coli + HE*	7.0 ± 1.7	30.0 ± 2.2	22.7 ± 2.2	40.2 ± 4.6

**Table 2 biomolecules-13-00098-t002:** Solvent-accessible area parameters calculated from the full structure models of VEGF_165_ and VEGF_165_ + HE using PyMOL v. 1.6.

	Total Area, Å^2^	BSA, Å^2^	SAS, Å^2^	SASap, Å^2^	SASpol, Å^2^
VEGF_165_	39,586	17,422	22,166	5713	16,453
VEGF_165_ + HE	41,094	20,154	20,940	4856	11,254

SAS—solvent-accessible surface area; BSA—area buried from the solvent; SASap—solvent-accessible apolar area; SASpol—solvent-accessible polar area.

## Data Availability

Not applicable.

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
