# Peer review of "Heparin-Induced Changes of Vascular Endothelial Growth Factor (VEGF165) Structure"

_biomolecules, 2023, doi:10.3390/biom13010098_

Round 1
Reviewer 1 Report
1. In Table 1, The CD data of rhVEGF165-E.coli with heparin should be included.
2. In Figure 2B, the blue peak at ~1000nm should be explained.
3. The biological meanings of following terms should be discussed. SAA, total solvent accessible area, BSA, area buried from the solvent, ASAap, solvent accessible apolar area, ASApol, solvent accessible polar area.
4. What is the heparin concentration to cause VEGF aggregation?
Reviewer 2 Report
Thank you for the opportunity to review your manuscript, "Heparin-induced Changes of Vascular Endothelial Growth Factor (VEGF-165) Structure." I found the content well-organized and well-researched, with strong arguments and an engaging discussion. However, there were a few areas where I had some suggestions for improvement. Overall, I believe that with these minor revisions, the manuscript has the potential to be accepted for publication. I hope my feedback is helpful, and I look forward to seeing the revised version.
Following are the comments on the manuscript.
Fluorescence spectroscopy studies. Section Results and discussion 4th Paragraph.
1. The observed red shift in the fluorescence spectrum of bis-ANS upon binding to rhVEGF165 in the presence of heparin is interesting and potentially informative. In general, a red shift in the fluorescence spectrum can be caused by an increase in the polarity of the environment surrounding the fluorophore, which can lead to a decrease in the strength of the non-covalent interactions between the fluorophore and the surrounding solvent or matrix. This can lead to a longer excited state lifetime and a shift in the emission spectrum towards longer wavelengths. In this study, the red shift observed upon heparin binding to rhVEGF165 may be due to a change in the environment surrounding the fluorophore due to the binding of heparin to the protein.
2. Here is the follow-up comment on the same section: As the authors haven’t mentioned anything about the inner filter effect in the methods, I assume it’s not included as part of the experiment. So based on this: It is important to consider the potential impact of the inner filter effect on these results. The inner filter effect refers to the absorption of light by the solvent or sample matrix, which can affect the measurement of fluorescence intensity. Suppose the solvent or sample matrix absorbs a significant amount of light. In that case, it can cause a reduction in the measured fluorescence intensity and potentially impact the interpretation of the red shift observed in the spectrum. It would be helpful to consider the inner filter effect and how it may have affected the measurement of the red shift in this study. Also please include a detailed methodology not just give the reference.Please justify! above two comments
3. Section 3.1: Overall, these findings provide valuable information about the secondary structure of VEGF165 and the effect of HE on this structure. However, it would be helpful to provide more context and to discuss the potential implications of these results in relation to the function and regulation of the protein. It is also unclear why the results obtained with eukaryotic and prokaryotic cells are mentioned and how these findings compare to previous studies on the secondary structure of VEGF165. Please justify.
Round 2
Reviewer 1 Report
The revised MS has addressed all my questions and can be accepted for publication.